perinatal depression; remission; untreated depression; India; Pakistan

**Corresponding authors:** Daniela C. Fuhr;
Email: fuhr@leibniz-bips.de
Benedict Weobong;
Email: bweobong@ug.edu.gh

# Predictors of spontaneous remission and recovery among women with untreated perinatal depression in India and Pakistan

Daniela C. Fuhr[1,2,3] , Siham Sikander[4,5], Fiona Vanobberghen[6,7,8],
Benedict Weobong[9], Atif Rahman[5] and Helen A. Weiss[8]

[1]Department of Prevention and Evaluation, Leibniz Institute of Prevention Research and Epidemiology, Bremen, Germany; [2]Health Sciences, University of Bremen, Bremen, Germany; [3]Department of Health Services and Policy Research, Faculty of Public Health and Policy, London School of Hygiene and Tropical Medicine, London, UK; [4]Human Development Research Foundation, Rawalpindi, Pakistan; [5]Department of Primary Care and Mental Health, University of Liverpool, Liverpool, UK; [6]Department of Medicine, Swiss Tropical and Public Health Institute, Allschwil, Switzerland; [7]University of Basel, Basel, Switzerland; [8]MRC International Statistics and Epidemiology Group, Faculty of Epidemiology and Population Health, London School of Hygiene and Tropical Medicine, London, UK and [9]Department of Social and Behavioural Sciences, College of Health Sciences, University of Ghana, Accra, Ghana

## Abstract

**Background:** Mothers with perinatal depression can show different symptom trajectories and may spontaneously remit from depression, however, the latter is poorly understood. This is the first study which sought to investigate predictors of spontaneous remission and longer-term recovery among untreated women with perinatal depression.

**Methods:** We analysed data from two randomised controlled trials in women with perinatal depression in India and Pakistan. Analyses were restricted to women in the control groups who did not receive active treatment. Generalised estimating equations and logistic regressions were used to estimate odds ratios (ORs) and 95% confidence intervals (CIs), adjusting for within-person correlation.

**Results:** In multivariable analyses, remission was associated with a husband who is not working (adjusted OR, aOR = 2.04, 95% CI 1.02–4.11), lower Patient Health Questionnaire-9 score at baseline (aOR = 0.43, 95% CI 0.20–0.90 for score of ≥20 vs. 10–14) and better social support at baseline (aOR = 2.37, 95% CI 1.32–4.27 for high vs. low social support).

**Conclusions:** Women with low baseline severity may remit from perinatal depression with adequate social support from family and friends. These factors are important contributors to the management of perinatal depression and the prevention of clinical worsening, and should be considered when designing low-threshold psychological interventions.

## Impact statement

Perinatal depression is a disabling mental condition that can significantly affect mothers after child birth and within the first year postpartum. The majority of women who develop depression after birth may show mild or moderate symptoms only, that have the potential to resolve over time without providing treatment. However, there is paucity of evidence of what may trigger spontaneous remission of perinatal depression and which factors are associated with this. With this study, we shed light on factors that may determine remission and recovery of symptoms. We do this by analysing data of mothers with depression who did not receive active treatment for their symptoms and followed them up over 6 months' time.

## Introduction

Perinatal depression is a disabling mental condition that has its onset during pregnancy or within the first year after delivery, and can significantly impair psycho-social functioning (DSM-V, 2013; Woody et al., 2017). In a recent systematic review and meta-analysis, the estimated prevalence of perinatal depression globally was higher among women in low- and middle-income countries (18.7% postnatally and 19.2% prenatally) than among women in high income countries (9.5% postnatally and 9.2% prenatally) (Woody et al., 2017). Mild and moderate symptoms of depression are most common (Hegel et al., 2006), however, for perinatal depression specifically, heterogeneity of symptoms has been reported (Santos et al., 2017). Low, medium and chronic symptom trajectories for perinatal depression have been established (Santos et al., 2017) but these may not necessarily follow expected patterns and may be transient (Baron et al., 2017). It is speculated that symptom trajectories may be influenced by specific biological, social and

psychological risk factors. These include prior history of psycho-pathology, high parity, low social support, lack of education and exposure to negative life events (Yim et al., 2015; Ahmed et al., 2019). However, we currently lack knowledge about modifiable risk factors and these still need to be established (Baron et al., 2017). The heterogeneity of symptoms of perinatal depression led to a call to personalise treatment to improve the response to treatment and facilitate recovery (Johansen et al., 2019). Our previous research on the effectiveness of brief psychological interventions for women with perinatal depression supports this tailored approach to treatment and suggests that a low-intensity psychological intervention for perinatal depression delivered by peers may be most suitable for women whose depression is mild and acute rather than chronic and severe (Fuhr et al., 2019; Sikander et al., 2019; Singla et al., 2021).

Research also reports spontaneous remission for perinatal depression (Yazici et al., 2015; Maselko et al., 2020). Spontaneous remission is attained when sub-clinical levels of symptoms of depression are achieved without treatment, determined for example by a below-threshold score on a standardised symptom severity measure (Whiteford et al., 2013). It has been more extensively studied for major depressive disorder, and a systematic review found that among people with untreated depression, 53% had spontaneous remission within 1 year (Whiteford et al., 2013). Spontaneous remission may be associated with the presence of other mental health problems, the duration of the overall episode and severity of symptoms, individual genetic vulnerabilities, personality traits as well as environmental supports and stressors (Paykel, 1998; Rush et al., 2006). It is especially common among people who show mild or moderate symptoms of depression only (Posternak and Miller, 2001; Hegel et al., 2006), and this has been confirmed for pregnant and post-partum women as well (Yazici et al., 2015; Santos et al., 2017; Maselko et al., 2020). Although there is evidence that women with perinatal depression may show spontaneous remission, and improve symptoms over time, the predictors for spontaneous remission and recovery (defined as remission at both 3 and 6 months) still need to be investigated further. This will strengthen the evidence base for the prevention and treatment of perinatal depression, and increase our understanding of the contribution of active treatment to remission and recovery.

In this study, we conducted secondary analyses of data from two completed randomised controlled trials (RCTs) in India and Pakistan which demonstrated the effectiveness of a peer-delivered psychological intervention on symptoms of perinatal depression (Fuhr et al., 2019; Sikander et al., 2019). We analysed data from women with perinatal depression in the control groups of the two trials who did not receive active treatment. The objectives of this study were a) to understand factors associated with remission among women with untreated perinatal depression in India and Pakistan; b) to explore factors associated with longer-term recovery; and c) to investigate if these factors differ by country.

## Methods

We pooled data from two RCTs which sought to investigate the effect of a task-shifted psychological intervention (The Thinking Healthy Programme-Peer delivered [THPP]) on the severity of symptoms for perinatal depression and remission (Vanobberghen et al., 2020). The results from the trials (an individually randomised trial in urban Goa, India; and a cluster RCT in rural sub-districts of Rawalpindi, Pakistan) have been published previously (Fuhr et al., 2019; Sikander et al., 2019). Data from these trials were collected during 2014–2016. Participants were recruited from antenatal centres and primary health care in Goa, and from community-based households in Rawalpindi. In both countries, participants were aged ≥18 years, in their second or third trimester of pregnancy at enrolment, and scored ≥10 on the Patient Health Questionnaire-9 (PHQ-9). The intervention was delivered by peers who were women from the local community who had children themselves and had gone through the experience of pregnancy, childbirth and raising a family.

Women in the control group received enhanced usual care. This included the following: First, women and their antenatal health care providers (antenatal health care providers were gynaecologists in Goa, and local government employed community health workers in Rawalpindi) were informed about the PHQ-9 score and depression status of participants. We also provided referral information to the gynaecologist (in Goa) and to the primary care physician (in Pakistan) who received training on the mental health gap intervention guide and referrals. Second, participants were provided with an information leaflet containing information about self-care during pregnancy and local self-referral pathways to mental health care. In addition to enhanced usual care, women in the intervention group received the THPP, a task-shifted psychological intervention based on behavioural activation which was delivered in 6–14 sessions over 6 months.

The primary trial outcome was remission, defined as the absence of depressive symptoms (PHQ-9 < 5) at 3 months post birth. The PHQ-9 was translated and validated in the local languages. Secondary outcomes included remission at 6 months, and recovery (PHQ-9 < 5 at both 3 and 6 months post birth).

Data were collected on the following baseline characteristics of the mother: Age in years; level of education (no formal education, up to primary education, up to secondary education, beyond secondary education); employment status (employed or not being currently employed); total number of children; previous miscarriage or still birth (none, one or more); sex of the current baby (girl or boy); family structure (nuclear or extended, i.e., living on their own with family or extended families living together); financial empowerment (yes/no), that is, asking if mother is able to put aside money for their own personal use or not; debt (yes/no), that is, asking if anyone in household is currently in debt or not; and if the mother has experienced any domestic violence in the past 3 months (yes/no). Perceived social support of the mother was measured with the Multidimensional Scale of Perceived Social Support. We also asked the mother about her expectation and how she would find the THPP intervention prior to starting it (not/little useful, somewhat useful, moderately useful, very useful). We collected data on her husband, namely, his level of education (no formal education, up to primary education, up to secondary education, beyond secondary education), employment status (employed or not being currently employed) and the number of months the husband had been away from home in the past 6 months (less than 1 month, 1 month or more).

For this paper, analyses were restricted to participants randomised to the control group of the trials. Data analysis was conducted using Stata version 17.0 (Stata, 2022). We used generalised estimating equations (GEEs) to adjust for within person correlation, and estimated odds ratios (ORs) and 95% confidence intervals (CIs) using logistic regression. Factors associated with remission were estimated by conducting a repeat measures analysis using data from both 3- and 6-month visits for each woman. We adjusted for country and month of visit (a priori), and factors associated with missing outcome data ('minimally adjusted

model'). We performed complete case analyses, that is, visits were included if remission data were available for that visit. We conducted sensitivity analyses using multiple imputation as an alternative strategy to account for missing outcome data. This imputed missing outcome data with 25 imputations. The final multivariable model included variables independently associated with the outcome, determined by starting with a full model of variables associated with the outcome with $p < 0.1$ in the minimally adjusted model, and retaining them if they were independently associated with the outcome in the multivariable model ($p < 0.05$). Country was an a priori effect-modifier, and effect-modification by country was assessed for the final models by fitting an interaction term of country and each exposure variable.

The study was performed in accordance with the ethical standards as laid down in the Declaration of Helsinki and its later amendments. Ethical approval was obtained from the Institutional Review Boards at the London School of Hygiene and Tropical Medicine, Sangath (the implementing institution in India), the Indian Council of Medical Research, the University of Liverpool and the Human Development Research Foundation (the implementing organisation in Pakistan). All study participants provided their informed consent prior to their inclusion in the study.

## Results

In India, 118,260 women were assessed for eligibility over a 2-year period, of whom 6,369 were screened using the PHQ-9, and 333 were eligible. Of these, 280 (84.1%) women were enrolled into the trial (140 randomly allocated per group). Over the same period, 1,910 pregnant women in Pakistan from 40 village clusters were assessed. Of these, 1,731 were screened using the PHQ-9, and 572 were eligible. Of these 570 (99%) were enrolled into the trial (283 in the intervention group and 287 in the control group).

Baseline characteristics of the 427 participants randomised to the control groups are shown in Table 1. Participants in India were slightly younger than those in Pakistan (mean 25.3 and 27.3 years, respectively) and had less education overall (55.7% of those in India had no formal education or primary education only, compared with 26.9% in Pakistan). The majority of participants in both countries did not work, however, participants in India were less likely to be in debt and reported being more financially empowered than those in Pakistan (78.6% vs. 54.7%). A higher proportion of participants in India had fewer than two children, and reported fewer miscarriages or stillbirths than participants in Pakistan. Participants in India tended to have less severe depression than those in Pakistan (71.4% with moderate depression in India vs. 58.2% in Pakistan), and to have had depression for a shorter period (35.7% with chronicity ≥12 weeks at baseline assessment vs. 80.3% in Pakistan). Participants in India reported higher perceived social support but had lower expectations of the usefulness of psychological counselling.

Overall, 56 (13.1%) of participants were not seen at both 3 and 6 months (i.e., lost to follow-up). Of these, 9 participants were in India (6.4% loss to follow-up), and 47 were in Pakistan (16.5% loss to follow-up). Factors independently associated with loss to follow-up were country ($p = 0.003$), and whether the woman worked outside the home ($p = 0.01$). These were adjusted for in subsequent analyses.

The prevalence of remission at 3 months and 6 months, respectively, were each lower among participants in Pakistan than those in India (44.1% vs. 50.8% at 3 months; 44.7% vs. 59.7% at 6 months; Appendix 1 of the Supplementary Material). This difference was

**Table 1.** Baseline characteristics of participants with untreated perinatal depression

| | India (N = 140) | Pakistan (N = 287) |
|---|---|---|
| Age, years of mother (mean (SD; range)) | 25.3 (4.74) (18–39) | 27.3 (4.97) (18–45) |
| Sex of current baby (n (%)) | | |
| Girl | 63 (48.1%) | 114 (47.3%) |
| Boy | 68 (51.2%) | 127 (52.7%) |
| Missing | 9 | 46 |
| Level of education (mother) (n (%)) | | |
| No formal education | 13 (9.3%) | 55 (19.2%) |
| Up to primary | 65 (46.4%) | 22 (7.7%) |
| Up to secondary | 46 (32.9%) | 162 (56.5%) |
| Beyond secondary | 16 (11.4%) | 48 (16.7%) |
| Occupation (mother) (n (%)) | | |
| Not employed | 118 (84.3%) | 270 (94.1%) |
| Employed | 22 (15.7%) | 17 (5.9%) |
| Level of education (husband) (n (%)) | | |
| No formal education/up to primary | 66 (47.1%) | 26 (9.1%) |
| Up to secondary | 52 (37.1%) | 222 (77.4%) |
| Beyond secondary | 22 (15.7%) | 39 (13.6%) |
| Occupation (husband) (n (%)) | | |
| Employed | 138 (98.6%) | 258 (89.9%) |
| Not employed | 2 (1.4%) | 29 (10.1%) |
| Husband away from home during past 6 months (n (%)) | | |
| <1 month | 127 (90.7%) | 200 (77.5%) |
| ≥1 month | 13 (9.3%) | 58 (22.5%) |
| Total number of children (n (%)) | | |
| 0 | 65 (46.4%) | 54 (20.2%) |
| 1 | 52 (37.1%) | 82 (30.7%) |
| ≥2 | 23 (16.4%) | 131 (49.1%) |
| Family structure (n (%)) | | |
| Nuclear family | 63 (45.0%) | 71 (24.7%) |
| Extended families living together | 77 (55.0%) | 216 (75.5%) |
| Debt of household (n (%)) | | |
| No | 83 (59.3%) | 118 (41.1%) |
| Yes | 57 (40.7%) | 169 (58.9%) |
| Financial empowerment of mother (n (%)) | | |
| No | 30 (21.4%) | 130 (45.3%) |
| Yes | 110 (78.6%) | 157 (54.7%) |
| PHQ-9 score (median (IQR)) | 12 (11–15) | 14 (12–17) |
| PHQ-9 category (n (%)) | | |
| Moderate (score 10–14) | 100 (71.4%) | 167 (58.2%) |
| Moderately severe (15–19) | 35 (25.0%) | 88 (30.7%) |
| Severe (20–27) | 5 (3.6%) | 32 (11.2%) |

*(Continued)*

**Table 1.** (*Continued*)

|  | India (*N* = 140) | Pakistan (*N* = 287) |
|---|---|---|
| Chronicity of depression, weeks (*n* (%)) | | |
| <12 weeks | 90 (64.3%) | 38 (19.7%) |
| ≥12 weeks | 50 (35.7%) | 155 (80.3%) |
| Missing | 0 | 94 |
| MSPSS score (mean (SD)) | 5.40 (1.10) | 3.95 (1.33) |
| Participant's expectation of usefulness of counselling (*n* (%)) | | |
| Not/a little useful | 30 (21.4%) | 15 (5.2%) |
| Somewhat useful | 32 (22.9%) | 60 (21.0%) |
| Moderately useful | 28 (20.0%) | 124 (43.4%) |
| Very useful | 50 (35.7%) | 87 (30.4%) |
| Parity (n (%)) | | |
| Primiparous | 58 (41.4%) | 50 (17.4%) |
| Multiparous | 82 (58.6%) | 237 (82.6%) |
| Previous miscarriage or still birth (*n* (%)) | | |
| None | 130 (92.9%) | 185 (64.5%) |
| One/more | 10 (7.1%) | 102 (35.5%) |
| Any domestic violence in last 3 months (*n* (%)) | | |
| No | 119 (85.0%) | 241 (85.5%) |
| Yes | 21 (15.0%) | 41 (14.6%) |

Abbreviation: MSPSS, Multidimensional Scale of Perceived Social Support.

statistically significant (minimally adjusted OR [aOR] = 1.55; 95% CI 1.08–2.22; *p* = 0.02; for the two endpoints combined; Table 2).

The association between other factors and remission are presented in Table 2.

In the multivariable model (Table 3), remission was more likely among participants with a husband who was not working (aOR = 2.04, 95% CI 1.02–4.11 for not working vs. working), lower PHQ-9 score at baseline (aOR = 0.43, 95% CI 0.20–0.90 for score of ≥20 vs. 10–14) and better social support at baseline (aOR = 2.41, 95% CI 1.35–4.31 for high vs. low social support). There was no evidence of a difference in remission by country after adjusting for these factors (*p* = 0.28). The results from multiple imputation gave similar results (Table 3). There was strong evidence of a greater effect of high social support versus low/moderate support on remission in India than in Pakistan (*p* = 0.003), and strong evidence of a trend with greater social support in India (*p* = 0.003) (Appendix 2 of the Supplementary Material).

Overall, 317 participants had data at both 3- and 6-month time points, and of these 100 (31.6%) had recovery. In multivariable analyses, improved recovery was associated with country (aOR = 1.81, 95% CI 1.02–3.19, *p* = 0.04 for India vs. Pakistan), and to a lesser extent with better social support (aOR = 2.03, 95% CI 0.84–4.89 for high vs. low social support; *p*-value for trend = 0.08), higher husband's education (aOR = 2.22, 95% CI 0.98–5.03; *p*-value for trend = 0.05) and lower PHQ-9 score (aOR = 0.36, 95% CI 0.10–1.29, *p*-value for trend = 0.09). Power was relatively lower for this analysis. With 317 participants with data on recovery, we had 80% power to detect a difference in proportions of 25% vs. 40% for variables where 50% of the study population are exposed. Data

on recovery are presented in Appendix 3 of the Supplementary Material.

## Discussion

To the best of our knowledge, this is the first study which assesses predictors of spontaneous remission and recovery in women with untreated perinatal depression. Almost half the women who did not receive active treatment for perinatal depression had spontaneous remission at 3 or 6 months post birth. Baseline factors associated with remission were lower baseline severity, strong social support from family and friends, and a husband who was not working. Recovery was lower than remission with around a third of women showing spontaneous remission at both 3 and 6 months post birth. Power was lower to detect associations with recovery but there were indications of similar factors as for remission. Remission and recovery were more common in India than in Pakistan in univariable analyses, with evidence of improved remission with greater social support in India.

The finding that participants with lower baseline symptom severity are more likely to remit is similar to research on major depressive disorder (Keller et al., 1992; Paykel, 1998; Posternak and Miller, 2001; Rush et al., 2006). Social support, in the form of instrumental and emotional help has also been identified as an important factor influencing spontaneous remission for depression (Hegel et al., 2006; Fuller-Thomson et al., 2014). Qualitative research further substantiates this finding and confirms that women with depression need a reliable social support system to facilitate the process of natural recovery (Naeem et al., 2004). A mother with a husband who is not working was also associated with remission in our analyses. One could speculate that this might infer social support being provided, however, unemployment of the husband may also strain the financial situation of the household and may lead to stress and insecurities in the household. This may aggravate mental health symptoms as seen in other studies (Patel et al., 2002). Therefore, this finding should be interpreted with caution and unpacked further in future studies. Strengths of our study include the size and strong internal validity of the trial procedures (such as no differential drop out, use of adequate control groups, and adherence to research protocols) but our study has also some limitations. First, we were reliant on the baseline variables of the trials and did not have measures of some factors for spontaneous remission and recovery which have been identified for major depressive disorder (Rush et al., 2006; Fuller-Thomson et al., 2014). These include the presence of other mental health problems, personality disorders or physical health problems. Whilst we screened for serious mental health problems such as psychosis, and excluded women who needed acute medical help, we did not screen for personality disorders. Personality traits might hinder spontaneous remission, and there is evidence suggesting a negative association between passive dependent personality traits and remission (Paykel, 1998). The type of coping and how to generally deal with life problems may also be important. Problem-focused coping has been shown to promote spontaneous remission, whereas the use of avoidant coping mechanisms may leave people feeling stuck in a specific situation, not allowing them to experience reinforcing activities (Hegel et al., 2006).

Second, there may be effects of being enrolled in a trial which may have led to the improvement of symptoms and to spontaneous remission of perinatal depression in our study. Evidence shows that follow-up assessments (Posternak and Zimmerman,

**Table 2.** Factors associated with remission (PHQ < 5 at 3 or 6 months)

| | N (3 m) | Remission at 3 m n (%) | N (6 m) | Remission at 6 m n (%) | OR for remission* (95% CI) | p-Value* |
|---|---|---|---|---|---|---|
| Country | | | | | | |
| Pakistan | 211 | 93 (44.1%) | 226 | 101 (44.7%) | 1 | 0.02 |
| India | 122 | 62 (50.8%) | 129 | 77 (59.7%) | 1.55 (1.08–2.23) | |
| Age group, years | | | | | | |
| ≤24 | 126 | 59 (46.8%) | 135 | 74 (54.8%) | 1 | 0.82 |
| 25–29 | 105 | 48 (45.7%) | 111 | 49 (44.1%) | 0.92 (0.60–1.40) | |
| ≥30 | 102 | 48 (47.1%) | 109 | 55 (50.5%) | 1.05 (0.69–1.61) | |
| Sex of current baby | | | | | | |
| Male | 156 | 72 (46.2%) | 168 | 82 (48.8%) | 1 | 0.66 |
| Female | 177 | 83 (46.9%) | 186 | 96 (51.6%) | 1.08 (0.77–1.52) | |
| Level of education (mother) | | | | | | |
| No formal education | 46 | 14 (30.4%) | 56 | 28 (50.0%) | 1 | 0.67 |
| Up to primary | 75 | 34 (45.3%) | 80 | 46 (57.5%) | 1.16 (0.63–2.13) | |
| Up to secondary | 166 | 84 (50.6%) | 172 | 80 (46.5%) | 1.30 (0.78–2.15) | |
| Beyond secondary | 46 | 23 (50.0%) | 47 | 24 (51.1%) | 1.43 (0.75–2.72) | |
| Occupation (mother) | | | | | | |
| Employed | 28 | 14 (50.0%) | 29 | 13 (44.8%) | 1 | 0.58 |
| Not employed | 305 | 141 (46.2%) | 326 | 165 (50.6%) | 1.20 (0.64–2.25) | |
| Level of education (husband) | | | | | | |
| No formal education/up to primary | 73 | 27 (37.0%) | 81 | 43 (53.1%) | 1 | 0.06 |
| Up to secondary | 209 | 98 (46.9%) | 221 | 106 (48.0%) | 1.55 (0.95–2.51) | |
| Beyond secondary | 51 | 30 (58.8%) | 53 | 29 (54.7%) | 2.05 (1.12–3.75) | |
| Occupation (husband) | | | | | | |
| Employed | 310 | 143 (46.1%) | 330 | 162 (49.1%) | 1 | 0.06 |
| Not employed | 23 | 12 (52.2%) | 25 | 16 (64.0%) | 1.93(0.97–3.84) | |
| Months husband away from home in last 6 months | | | | | | |
| 0 | 257 | 117 (45.5%) | 274 | 140 (51.1%) | 1 | 0.76 |
| ≥1 month | 54 | 26 (48.1%) | 57 | 22 (38.6%) | 0.93 (0.58–1.50) | |
| Total number of children | | | | | | |
| 0 | 94 | 45 (47.9%) | 100 | 54 (54.0%) | 1 | 0.87 |
| 1 | 106 | 48 (45.3%) | 117 | 58 (49.6%) | 0.93 (0.60–1.45) | |
| ≥2 | 120 | 55 (45.8%) | 125 | 61 (48.8%) | 1.04 (0.66–1.65) | |
| Family structure | | | | | | |
| Nuclear | 104 | 47 (45.2%) | 106 | 54 (50.9%) | 1 | 0.51 |
| Extended | 229 | 108 (47.2%) | 249 | 124 (49.8%) | 1.14 (0.78–1.68) | |
| Debt | | | | | | |
| Yes | 168 | 69 (41.1%) | 192 | 91 (47.4%) | 1 | 0.13 |
| No | 165 | 86 (52.1%) | 163 | 87 (53.4%) | 1.30 (0.92–1.85) | |
| Financial empowerment | | | | | | |
| No | 118 | 46 (39.0%) | 128 | 55 (43.0%) | 1 | 0.03 |
| Yes | 215 | 109 (50.7%) | 227 | 123 (54.2%) | 1.48 (1.03–2.14) | |
| Chronicity of depression, weeks | | | | | | |
| <12 | 108 | 51 (47.2%) | 105 | 55 (52.4%) | 1 | 0.60 |
| ≥12 | 151 | 63 (41.8%) | 172 | 79 (45.9%) | 1.13 (0.72–1.77) | |

*(Continued)*

**Table 2.** (*Continued*)

|  | N (3 m) | Remission at 3 m n (%) | N (6 m) | Remission at 6 m n (%) | OR for remission* (95% CI) | p-Value* |
|---|---|---|---|---|---|---|
| PHQ-9 category |  |  |  |  |  |  |
| 10–14 | 209 | 100 (47.9%) | 223 | 124 (55.6%) | 1 | 0.02 |
| 15–19 | 99 | 50 (50.5%) | 105 | 45 (42.9%) | 0.83 (0.57–1.21) |  |
| ≥20 | 25 | 5 (20.0%) | 27 | 9 (33.3%) | 0.37 (0.18–0.76) |  |
| MSPSS category |  |  |  |  |  |  |
| Low | 51 | 17 (33.3%) | 58 | 19 (32.8%) | 1 | 0.006 |
| Moderate | 164 | 70 (42.7%) | 176 | 89 (50.6%) | 1.72 (1.03–2.88) |  |
| High | 118 | 68 (57.6%) | 121 | 70 (57.9%) | 2.53 (1.42–4.50) |  |
| Participant's expectation of usefulness of counselling |  |  |  |  |  |  |
| Not/a little useful | 39 | 17 (43.6%) | 38 | 19 (50.0%) | 1 | 0.88 |
| Somewhat useful | 70 | 33 (47.1%) | 76 | 39 (51.3%) | 1.23 (0.65–2.33) |  |
| Moderately useful | 116 | 55 (47.4%) | 123 | 58 (47.2%) | 1.26 (0.69–2.33) |  |
| Very useful | 108 | 50 (46.3%) | 118 | 62 (52.5%) | 1.28 (0.70–2.33) |  |
| Parity |  |  |  |  |  |  |
| Primiparous | 84 | 42 (50.0%) | 91 | 52 (57.1%) | 1 | 0.36 |
| Multiparous | 249 | 113 (45.4%) | 264 | 126 (47.7%) | 1.21 (0.81–1.81) |  |
| Previous miscarriage or still birth |  |  |  |  |  |  |
| None | 244 | 118 (48.4%) | 263 | 137 (52.1%) | 1 | 0.48 |
| One or more | 89 | 37 (41.6%) | 92 | 41 (44.6%) | 0.86 (0.57–1.30) |  |
| Domestic violence |  |  |  |  |  |  |
| No | 284 | 137 (48.2%) | 297 | 155 (52.2%) | 1 | 0.07 |
| Yes | 45 | 18 (40.0%) | 53 | 21 (39.6%) | 0.64 (0.39–1.04) |  |

Abbreviation: MSPSS, Multidimensional Scale of Perceived Social Support.
*Using both 3- and 6-month data, assuming a constant association of exposure at these two time points, and adjusted for month of visit and factors associated with loss to follow-up (namely, country and women's occupation).

2007), and informing the patient about their clinical status and illness (Maselko et al., 2020) may lead to the improvement of outcomes. This may be explained by the installation of hope, the positive expectation from receiving guidance or therapeutic assessments (Posternak and Miller, 2001). Other authors (Hengartner, 2020) substantiate this and explain symptom reduction in depression trials by the Hawthorne effect resulting from receiving any care or additional support such as enhanced usual care. Regression to the mean is another plausible explanation of spontaneous remission and recovery for depression (Hengartner, 2020) which might have contributed to the high prevalence of remission in our study. Furthermore, we did not conduct clinical interviews with women, but measured symptom severity with the PHQ-9 scale. Symptom scales have been shown to overestimate the prevalence of depression (Thombs et al., 2018), and may have facilitated the inclusion of women with low baseline severity. For perinatal depression specifically, there have been calls to suggest that the assessment of symptoms lasting 2 weeks may not be sufficient to exclude spontaneous remission without treatment (Scott, 1997). Inclusion of a high proportion of moderate cases of perinatal depression (overestimated by the screening tool) may have led in turn to the overestimation of remission rates in our study, given that cases with mild or moderate symptoms are more likely to remit (Whiteford et al., 2013). Third, our definition of

spontaneous remission and recovery did not include an assessment of psycho-social functioning and physiological factors (Whiteford et al., 2013) which at times is also considered in the definition of remission and recovery; this may have led to apparent higher remission and recovery rates in our sample.

## Conclusion

Women with perinatal depression represent a diverse sub-group, and it is important to identify modifiable factors which may offset symptom trajectories and facilitate the way to recovery. Our study has confirmed that spontaneous remission from depression is a common outcome among mothers. For the first time, we now know that factors such as less severe depression at baseline, and more social support from friends and family are predictive of spontaneous remission. For the purposes of management of depression and the prevention of clinical worsening, these factors can be screened at baseline to help assess treatment prognosis. We were unable to assess relapse of depressive symptoms and if recovery remained stable over the long term. This would have to be investigated in further research, alongside the assessment of relevance of these factors with regards to the specific symptom trajectory women might find themselves in. Future research should also explore the

**Table 3.** Factors independently associated with remission from depression

| | AOR for remission (95% CI)[1] | p-value | AOR for remission (95%CI)[1], using MI* | p-value |
|---|---|---|---|---|
| Country | | | | |
| Pakistan | 1 | 0.28 | 1 | 0.35 |
| India | 1.25 (0.84–1.86) | | 1.21 (0.81–1.80) | |
| Occupation (participant) | | | | |
| Employed | 1 | 0.51 | 1 | 0.62 |
| Not employed | 1.24 (0.65–2.35) | | 1.17 (0.63–2.17) | |
| Occupation (husband) | | | | |
| Employed | 1 | 0.04 | 1 | 0.08 |
| Not employed | 2.04 (1.02–4.11) | | 1.91 (0.94–3.91) | |
| PHQ-9 category | | | | |
| 10–14 | 1 | 0.04** | 1 | 0.03** |
| 15–19 | 0.87 (0.59–1.28) | | 0.84 (0.58–1.23) | |
| ≥20 | 0.43 (0.20–0.90) | | 0.47 (0.24–0.92) | |
| MSPSS category | | | | |
| Low | 1 | 0.003** | 1 | 0.003** |
| Moderate | 1.58 (0.93–2.67) | | 1.59 (0.98–2.59) | |
| High | 2.37 (1.32–4.27) | | 2.41 (1.35–4.29) | |

Abbreviation: MSPSS, Multidimensional Scale of Perceived Social Support.

[1]Adjusted for other variables in the table (country and women's occupation are included as they are associated with LTFU; month of visit is also adjusted for; other variables are included since $p < 0.05$ in the multivariable model).

*MI, multiple imputation for missing outcome data (imputed data at 166/854 data points).

**P-value for trend.

role of social support interventions in the management of perinatal depression.

**Open peer review.** To view the open peer review materials for this article, please visit http://doi.org/10.1017/gmh.2023.26.

**Supplementary material.** The supplementary material for this article can be found at https://doi.org/10.1017/gmh.2023.26.

**Data availability statement.** Data from our trials have been made available at the LSHTM data repository available at http://datacompass.lshtm.ac.uk/ (doi: 10.17037/DATA.00000793).

**Author contribution.** All authors contributed to the study conception and design. D.C.F. conceived the study and H.A.W. and F.V. analysed the data. The first draft of the manuscript was written by D.C.F. and all authors commented on previous versions of the manuscript. All authors read and approved the final version.

**Financial support.** The research reported in this publication was supported by the National Institute of Mental Health of the National Institutes of Health under award number 1U19MH095687. The content is solely the responsibility of the authors and does not necessarily represent the official views of the National Institute of Mental Health, the National Institutes of Health, or the U.S. Department of Health and Human Services. F.V. and H.A.W. were funded by the UK Medical Research Council (MRC) and the UK Department for International Development (DFID) under the MRC/DFID Concordat agreement which is part of the EDCTP2 programme supported by the European Union (Grant No. MR/R010161/1).

**Competing interest.** The authors declare that they have no competing interest.

**Ethics standard.** The study was performed in accordance with the ethical standards as laid down in the 1964 Declaration of Helsinki. Ethical approval was obtained from the institutional review boards at the London School of Hygiene and Tropical Medicine, Sangath (the implementing institution in India), the Indian Council of Medical Research, the University of Liverpool and the Human Development Research Foundation (the implementing organisation in Pakistan).

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
