## [Reviewer Report]

Dear Dixon Chibanda, dear Judy Bass, 

On behalf of the authors, I am submitting a paper entitled “Predictors of spontaneous remission and recovery among women with untreated perinatal depression in India and Pakistan”. 

This is the first study which investigates predictors for remission and recovery among a large group of untreated women with perinatal depression, and we analysed data from two completed randomised controlled trials in women with perinatal depression in India and Pakistan. 

Analyses were restricted to women in the control groups (who did not receive active treatment). Remission and recovery were measured with the Patient Health Questionnaire-9 (PHQ-9). Remission was defined as PHQ-9<5 at 3 or 6-months post-birth. Recovery was defined as PHQ-9 <5 at both 3- and 6-months post-birth. Generalised estimating equations and logistic regressions were used to estimate odds ratios (OR) and 95% confidence intervals (CI), adjusting for within-person correlation (combining 3- and 6-month data). 

The prevalence of remission at 3 months was slightly lower among the 287 participants in Pakistan than the 140 participants in India (44.1% versus 50.8%), with similar results at 6 months (44.7% versus 59.7%). In multivariable analyses, remission was associated with a husband who is not working (adjusted odds ratio, aOR=0.48, 95%CI 0.24-0.97 for working versus not working), lower PHQ-9 score at baseline (aOR=0.43, 95%CI 0.20-0.89 for score of >20 versus 10-14), and better social support at baseline (aOR=2.39, 95%CI 1.32-4.33 for high versus low social support). There was little evidence of an association of any factors with improved recovery. 

Our study confirms previous findings for major depressive disorder and shows that baseline severity, and social support from partners and family are important factors which help us understand spontaneous remission of women with perinatal depression. These predictors are important for intervention purposes and for the management of perinatal depression in women with mild or moderate perinatal depression. 

We would be delighted if our paper were published in Cambrige Prism: Global Mental Health. Please do not hesitate to get in touch with us if you require further information.

Sincerely, Daniela Fuhr 

Prof. Dr. Daniela Fuhr

Leibniz-Institut für Präventionsforschung und Epidemiologie - BIPS GmbH 

Abt. Prävention und Evaluation 

Leibniz Institute for Prevention Research and Epidemiology - BIPS GmbH 

Department of Prevention and Evaluation 

Achterstraße 30 

D-28359 Bremen 

Tel +49 421 21856 900 (secr.) / 21856 754 (dir.) 

Fax +49 421 21856 941 

E-Mail: fuhr@leibniz-bips.de 

www.leibniz-bips.de

Honorary Professor

London School of Hygiene and Tropical Medicine

---

## [Reviewer Report]

The authors examine factors associated with remission of perinatal depression in the absence of treatment. Control arms of RCTs conducted in India and Pakistan were used for this analysis. The topic as well as the setting underscores the importance of the study. Please see my comments below:

1. Throughout the manuscript, a number of sentences are very long involving “and” and “but”, sometimes multiple times within the same sentence. I recommend splitting those sentences up for more clarity and ease of reading.

2. Last sentence of page 2. “Although .....” is one such sentence. In addition, I am not sure if it is necessary to state, “the predictors of spontaneous remission and recovery <b>among women with untreated perinatal depression..</b>.” Doesn’t “spontaneous remission” automatically imply that the women received no treatment? It may be helpful to provide some kind of a definition for spontaneous remission at the start of this paragraph, even if that definition applies only in the context of this study.

4. Methods (page 4, line 120-121): please elaborate on “without outcome data”. Does this mean only those who had information for both 3 and 6-months follow up were included in the final analyses?

5. Please specify when (which year[s]) the trials were conducted.

6. Results, lines 151-154: how were moderate and severe depression defined? How was social support measured?

7. Given that the prevalence of remissions (lines 166-170) was statistically significantly different, it may be better to refrain from using “slightly” [line 166] and rather let the readers draw their own conclusions.

8. Lines 177-179: I think it makes more sense to follow the same “order” when presenting the adjusted OR’s within parentheses. For instance, having a husband who wasn’t working increased the likelihood of remission. Thus, it would be more intuitive to see the odds of remission for those with husbands not working vs working (the aOR within parentheses would be higher than 1), rather than the inverse.

9. Were the PHQ-9’s translated into local languages and were those already validated in local languages?

10. In this paper, it appears that husband not working is being used aas a surrogate measure for the level of support the woman may have at home. Is there evidence from elsewhere to support that? How might that interact with the financial stability of the household, in terms of their effect on perinatal depression?

11. I am wondering what was the P value cut off used to determine the variables for multivariable model? If it was 0.1, then based on table 2, other variables (such as husband level of education, domestic violence) deserve to be included. If it was 0.05, husband occupation need not be included. The limited description in methods does not make this clear.

---

## [Reviewer Report]

The current manuscript explores the predictors of remission of depression symptoms in the first 6 months post-partum among a control arm of a depression trial in India and Pakistan. The analysis is possible due to the relatively weak enhanced usual care received in this group and so this approach is unique. Below I outline a few changes that I believe would significantly improve the manuscript.

Sentence starting at line 40 in the introduction, what do the percentages refer to?

The methods section is missing details introducing the variables and measures/scales used in the analyses. For example, how was financial empowerment or domestic violence measured? This can also help avoid some confusion in table 1 where number of children is presented earlier in the table, while later there is information on parity and miscarriages.

It is not clear how the specific potential predictors of remission/recovery were chosen for consideration to be included in the model (besides country). Does the list reflect the full set of variables that were available at both locations?

In line 138, please clarify what is meant by 118,260 women being assessed for eligibility, what was the process? The initial numbers are drastically different between India and Pakistan, suggesting a different recruitment strategy.

On line 146, the comparison of education could be made much clearer by either stating ‘fewer than secondary’ or acknowledging the bimodal difference in that, in Pakistan, a higher proportion of women either had no education or had higher levels of education than in India.

Please define what does ‘combining data from both 3 and 6 months’ mean in the analyses– is it remission at either 3 or 6 months?

Starting on line 177, the text presents the table 3 results in a confusing way, presenting the flip of the presented regression results. It would be easier for the reader to state, for example, that husband working and higher phq-9 scores were associated with a lower odds of remission, while higher social support was associated with greater odds of remission.

The paragraph on recovery is presented more like an after-thought. For example, no information is given about the significantly reduced sample size. These results would seem more important to present than, for example, the MI results which are currently in table 3. This is especially so given that the results on recovery are important enough to be summarized in the first paragraph of the discussion. On the other hand, the authors do not return to a discussion about how there do not seem to be any strong predictors of recovery, which seems like a lost opportunity.

The first paragraph of the discussion mentions adjustment for confounders but the analysis does not appear to be driven at all by concerns of confounding and/or causality.

The discussion misses several opportunities to elaborate on the potential reasons underlying the findings – especially things like husband working (i.e. what does this mean in terms of SES?), and the fact that the strong initial predictive value of country seems to weaken in the final model.

---

## [Reviewer Report]

Dear authors,

This study is of interest to reader of this journal.Howerver, both reviewers raised significant concerns about the conceptual frame work, methods of the study and the quality of the writing. Nevertheless, I am willing to give you the opportunity to revise your paper and respond to each of their comments.

---

## [Reviewer Report]

Based on the authors' responses, I believe my concerns to the initial draft have been addressed. I have no further comments.